# Molecular Characterization and Functional Analysis of Three Autophagy Genes, *BxATG5*, *BxATG9*, and *BxATG16*, in *Bursaphelenchus xylophilus*

**DOI:** 10.3390/ijms20153769

**Published:** 2019-08-01

**Authors:** Hong-Bin Liu, Lin Rui, Ya-Qi Feng, Xiao-Qin Wu

**Affiliations:** 1Co-Innovation Center for Sustainable Forestry in Southern China, College of Forestry, Nanjing Forestry University, Nanjing 210037, China; 2Jiangsu Key Laboratory for Prevention and Management of Invasive Species, Nanjing Forestry University, Nanjing 210037, China

**Keywords:** pine wood nematode, autophagy, autophagy genes, RNA interference, expression patterns, α-pinene, H_2_O_2_

## Abstract

The pine wood nematode (PWN), *Bursaphelenchus xylophilus*, is the pathogen responsible for pine wilt disease (PWD), a devastating forest disease with a pathogenic mechanism that remains unclear. Autophagy plays a crucial role in physiological and pathological processes in eukaryotes, but its regulatory mechanism and significance in PWN are unknown. Therefore, we cloned and characterized three autophagy genes, *BxATG5*, *BxATG9*, and *BxATG16,* in PWN. *BxATG9* and *BxATG16* were efficiently silenced through RNA interference, and we found that *BxATG16* positively regulated the expression of *BxATG5*. Silencing *BxATG9* and *BxATG16* severely inhibited feeding and reproduction in PWN, indicating that autophagy is essential for these processes. We then examined the expression patterns of these three autophagy genes in PWN under the stresses of α-pinene and H_2_O_2_, the main defense substances of pine trees, and during the development of PWD using quantitative reverse transcription polymerase chain reaction. The expression levels of *BxATG5*, *BxATG9*, and *BxATG16* all significantly increased after nematodes were stressed with α-pinene and H_2_O_2_ and inoculated into pine trees, suggesting that autophagy plays an important role in the defense and pathogenesis of PWN. In this study, the molecular characteristics and functions of the autophagy genes *BxATG5*, *BxATG9*, and *BxATG16* in PWN were elucidated.

## 1. Introduction

The pine wood nematode (PWN), *Bursaphelenchus xylophilus*, is a widespread and important quarantine pest [1], and the causative agent of pine wilt disease (PWD), a devastating forest disease that causes enormous ecological and economic losses in many parts of Europe and Asia, particularly China [2,3,4,5]. Numerous studies have been conducted on PWN. Recently, the major sperm protein BxMSP10 was reported to be necessary for reproduction and egg hatching in PWN [6]. A comparative proteomic study of *Pinus massoniana* inoculated with PWN has been conducted [7], and research involving PWN control is continuing. A recent study found that the *Bacillus pumilus* strain LYMC-3 shows nematicidal activity against PWN [8]. However, PWD is highly complex, and the mechanisms through which PWN adapts to environmental changes and host defenses to efficiently infect pine trees have not been fully elucidated. Therefore, it remains very difficult to prevent and control PWD.

Autophagy involves the transportation of intracellular membranes to maintain cell physiology and is key to maintaining organismal health [9]. It can help cells respond to adverse conditions through degradation of intracellular components and is evolutionarily conserved across eukaryotes [10,11]. Autophagy has been reported to play essential roles in physiological and pathological processes, including starvation adaptation, development, anti-aging, and degradation of invading bacteria [12,13,14], and has become a hot topic of research in recent years. The process of autophagy is regulated by autophagy genes. Currently, researchers have identified 34 autophagy genes in *Saccharomyces cerevisiae*, and the proteins encoded by these genes are critical to the process of autophagy [15,16]. Previous research showed that autophagy genes affect the shape of dauer larvae, as well as the lifespan and survival rate of *Caenorhabditis elegans* during periods of starvation [12]. Autophagy genes have also been reported to play a significant role in the development, reproduction, and pathogenicity of pathogens and insect pests such as *Magnaporthe grisea*, *Rhipicephalus* (*Boophilus*) *microplus*, and *Tenebrio molitor* [17,18,19,20].

Our previous research using transmission electron microscopy (TEM) revealed that autophagy occurs in PWN during periods of starvation, and we subsequently cloned the autophagy genes *BxATG1*, *BxATG3*, *BxATG4*, *BxATG7*, and *BxATG8* from PWN through polymerase chain reaction (PCR). RNA interference (RNAi) in *BxATG1* and *BxATG8* was found to affect the reproduction and pathogenicity of PWN and autophagy also was proved to contribute to the mobility of PWN at low temperatures [21,22,23,24]. However, it remains unclear whether other autophagy genes are present in PWN. The autophagy genes *ATG5* and *ATG16* are core factors involved in autophagosome assembly in *S*. *cerevisiae*, and their proteins play major roles in a series of ubiquitin-like modifications that take place during autophagy. The autophagy protein ATG16 binds to the complex ATG12-ATG5 to form a large complex that acts to extend the membrane of autophagy vesicles and provides a platform for interactions of proteins involved in the formation of autophagy vesicles [25]. This step is essential to the process of autophagy. *ATG9* is the only autophagy gene that encodes a transmembrane protein, which is localized in the autophagic membrane or autophagy precursor in *S*. *cerevisiae* and may regulate autophagy by altering membrane trafficking. This ubiquitin-like protein may provide a membrane source for autophagic vacuole formation and is necessary for autophagosome membrane formation [26]. To investigate whether these important autophagy pathways are also present in PWN, we cloned and analyzed the homologs of three autophagy genes, *ATG5*, *ATG9*, and *ATG16*, in PWN.

Resin is the first line of defense against pathogen invasion in pine trees. It consists of monoterpenes, sesquiterpenes, and diterpenes, of which α-pinene is the main monoterpene [27]. Li et al. found that 56.33 mg/mL α-pinene severely inhibited the reproduction of PWN [28]. When plants are infected with pathogens, they produce a large amount of reactive oxygen species (ROS), which is one of the main defense responses of plants. ROS result from biological and cellular aerobic metabolism, with H_2_O_2_ being the most stable and having highest content [29]. When pine trees are infected with PWN, the resin and ROS contents increase greatly [5,30]. To determine whether autophagy is involved in the mechanism used by PWN to overcome pine defenses and successfully infect pine trees, we analyzed the expression patterns of the three aforementioned autophagy genes in PWN under α-pinene and H_2_O_2_ stress and during the development of PWD using quantitative reverse transcription PCR (qRT-PCR). The results provide valuable information about the defense mechanisms of PWN and pathogenesis of PWD.

## 2. Results

### 2.1. Cloning and Sequence Analysis of Three Autophagy Genes of PWN

To further investigate the autophagy mechanism of PWN, we analyzed the transcriptome and cloned multiple coding sequences (CDS) of autophagy genes from PWN using PCR. The PCR amplification bands produced for each of the three autophagy genes were single and bright (Figure 1A). A Blastx search of sequencing results showed that the proteins encoded by these three genes shared more than 40% homology with the autophagy proteins ATG5, ATG9, and ATG16 of some other nematodes, confirming that the three genes cloned from PWN were all autophagy genes. We named the genes BxATG5, BxATG9, and BxATG16 (accession numbers: MK387710, MK387711, and MK387712); they had CDS sizes of 762 bp, 2370 bp, and 1563 bp, respectively, encoding 253, 789, and 520 amino acids, respectively (Figure 1B).

Homology analysis of the PWN autophagy proteins BxATG5, BxATG9, and BxATG16 revealed that BxATG5 shared 53.45% homology with ATG5 from *Toxocara canis*, *Brugia malayi*, *Wuchereria bancrofti*, and *Spodoptera litura*; BxATG9 shared about 44.61% homology with ATG9 from *T. canis*, *B. malayi, W. bancrofti, Haemonchus contortus*, and *Dictyocaulus viviparus*; and BxATG16 shared about 46.73% homology with ATG16 from *Diploscapter pachys* and *C. elegans*. Interestingly, most of these nematode species are parasitic in humans. We constructed phylogenetic trees of these three proteins using the maximum likelihood method and found that they are not closely related to known autophagy proteins from other species (Figure 1C). We analyzed the physicochemical properties of these three proteins on the bioinformatics website ExPASy. The results showed that the molecular weights of BxATG5, BxATG9, and BxATG16 were 29.73, 90.59, and 59.07 kDa, respectively, with isoelectric points (pI) of 5.73, 5.35, and 5.06, respectively. In addition, the proteins were acidic and hydrophilic. We predicted their signal peptides and transmembrane and tertiary structures (Figure 1D), and found that BxATG5 and BxATG16 contained no signal peptides, whereas BxATG9 had a signal peptide and five transmembrane regions, indicating that BxATG9 is a typical transmembrane protein. Surprisingly, no spatial folding was found in the predictive model of the BxATG16 tertiary structure, which may be related to the characteristic function of the protein.

### 2.2. Silencing BxATG9 and BxATG16 Reduced PWN Feeding and Reproduction

To investigate whether autophagy genes affect the behavior of PWN, we silenced *BxATG9* and *BxATG16* through RNAi and then examined the feeding and reproduction of PWN. We efficiently silenced *BxATG9* and *BxATG16* by soaking PWN with dsBxATG9 and dsBxATG16, respectively, but failed to silence *BxATG5*. Surprisingly, we found that silencing *BxATG16* significantly reduced the expression level of *BxATG5*, whereas silencing *BxATG9* did not obviously affect the expression of the other genes (Figure 2A). We did not find similarity in the nucleotide sequences of *BxATG5* and *BxATG16*, but there was a close regulatory relationship between the two genes. Silencing those two genes did not affect the morphology or mobility of PWN. The effects of RNAi on feeding and reproduction of nematodes were tested using potato dextrose agar (PDA) plates inoculated with *B. cinerea* incubated at 25 °C. The results showed that silencing *BxATG9* and *BxATG16* both significantly reduced PWN feeding and reproductive rates. The reproductive rate of nematodes soaked in dsBxATG9 or dsBxATG16 was 0.39- or 0.44-fold that of nematodes soaked in dsGFP, respectively (Figure 2B,C). Thus, the three autophagy genes *BxATG5*, *BxATG9*, and *BxATG16* all play important roles in the feeding and reproduction of PWN.

### 2.3. α-pinene Stress Increased the Expression Levels of Autophagy Genes in PWN

α-pinene is one of the main defensive substances found in pine trees. To explore whether resistance to α-pinene is associated with autophagy genes in PWN, we examined the expression levels of autophagy genes in PWN under α-pinene stress. The expression levels of the autophagy genes *BxATG5*, *BxATG9*, and *BxATG16* all increased significantly when the nematode was stressed with α-pinene for 4 h, with 3.43-, 7.79-, and 4.19-fold changes from the control (Triton X-100), respectively, but expression levels did not change markedly with only 3 h of stress (Figure 3A–C). Since we did not find that the autophagy genes were downregulated during stress, we also tested the expression levels of the three autophagy genes when the nematode was stressed with α-pinene for 5 h, and found that they were downregulated to 0.89-, 1.27-, and 0.37-fold of the control value, respectively. These results indicate that α-pinene exposure is unlikely to directly affect the expression levels of autophagy genes, but autophagy may be a later response used by the PWN, and so these three autophagy genes are upregulated transiently at a later time point. At this point, the nematodes likely act to protect their cells by increasing the expression of autophagy genes to remove damaged proteins and organelles, meaning that autophagy genes may still play an important role in nematode resistance to resin in pine trees.

### 2.4. Response of Autophagy Genes in PWN to Oxidative Stress

PWN are subjected to oxidative stress during the early stage of infection in pine trees. To investigate whether autophagy genes are involved in the resistance of PWN to oxidative stress, we measured the expression levels of three autophagy genes in the nematodes under oxidative stress conditions. The results showed that the expression levels of *BxATG5*, *BxATG9*, and *BxATG16* increased significantly when the nematodes were stressed with H_2_O_2_ for 2, 1, and 4 h, respectively, showing 2.54-, 4.32-, and 2.72-fold variations from the control (ddH_2_O; Figure 4A–C). Our results suggest that PWN protect themselves by upregulating these genes to enhance autophagic activity and thus maintain physiological balance, and that the autophagy genes *BxATG5*, *BxATG9*, and *BxATG16* play different roles in regulating oxidative stress-induced autophagy.

### 2.5. Expression Patterns of Autophagy Genes in PWN during the Development of PWD

After identifying the important roles of the autophagy genes *BxATG5*, *BxATG9*, and *BxATG16* in the resistance of PWN to α-pinene and oxidative stress, we further tested the expression levels of these three genes during different stages of PWD and analyzed their expression patterns during PWD development. After two-year-old *P. massoniana* seedlings were inoculated with PWN, their needles began to appear yellow in color on the 12th day. On the 28th day, most of the needles had lost water and yellowed. On the 40th day, all of the needles were wilted, and the plants were dead (Figure 5A). The expression levels of the three investigated autophagy genes were all significantly elevated on days 6, 12, 28, and 40 after inoculation (Figure 5B–D). They all peaked at day 28, when they showed 8.22-, 8.64-, and 9.09-fold increases. These results indicate that autophagy genes play an important role in helping PWN to effectively infect and colonize pine hosts throughout the development of PWD. During the early stage of PWD, the PWN may resist the defense responses of pine trees by increasing the expression of autophagy genes. During the late stage of PWD, the expression levels of autophagy genes were upregulated to their peak levels, presumably because the number of nematodes in each pine tree was too high and food became scarce, and starvation conditions caused nematodes to produce more recyclable small biological molecules to survive by increasing their autophagic activity. A relative decrease in the expression levels of these three autophagy genes when the trees died likely occurred because PWN fed on the blue stain fungus that began to multiply rapidly in the dead trees.

## 3. Discussion

Autophagy is a response of eukaryotic cells to environmental stresses [31]. It is a protective mechanism used during the growth and development of eukaryotic cells to maintain metabolic balance and homeostasis by eliminating excess biological macromolecules and damaged organelles, and plays very important roles in cell waste removal, structural reconstruction, growth, and development [32,33]. Detailed research into autophagy and its role in the growth and pathogenesis of PWN is essential for elucidating the defense and pathogenic mechanisms of PWN, and thus can contribute to the effective control of PWD.

Autophagy is regulated by multiple autophagy genes, and the proteins encoded by these genes play key roles in the process of autophagy [16]. The regulatory mechanism of autophagy has been thoroughly studied in *S*. *cerevisiae* and mammals, and many studies have been conducted in *C. elegans*. Although autophagy is an evolutionarily conserved process, its molecular mechanism differs among species. Currently, only five autophagy genes from PWN, *BxATG1*, *BxATG3*, *BxATG4*, *BxATG7*, and *BxATG8,* have been reported [21,23]. To better understand the mechanism of autophagy regulation in PWN, we cloned and analyzed three new autophagy genes in PWN: *BxATG5*, *BxATG9*, and *BxATG16*. The proteins encoded by these genes were found to share some homology with autophagy proteins ATG5, ATG9, and ATG16, respectively, of some other nematodes, but their genetic relationships were not close. These three autophagy proteins are acidic and hydrophilic, indicating that the intracellular environment is likely acidic when autophagy is induced in the nematode. ATG16 is reported to bind to ATG5 and to be required for the function of the ATG12-ATG5 conjugate in the autophagy pathway in *S*. *cerevisiae*. ATG5 contains two ubiquitin-like domains flanking a helix-like structure, and the N-terminal region of ATG16 has a helical structure that acts in combination with the grooves formed by these three regions to expand the separator [25]. The C-terminal of ATG16 in mammals contains a repeating domain of WD amino acids, which makes ATG16 more conducive to protein–protein interactions [34]. BxATG16 in PWN also contains this domain, which presumably has a similar function to ATG16 in mammals. The tertiary structure model of BxATG16 showed no folding, which may be related to its specific function in autophagy. ATG9 in *S*. *cerevisiae* is reported to mediate the turnover of biomacromolecules that depend on vacuole degradation [26]. BxATG9 in PWN contains a signal peptide and five transmembrane regions, indicating that it is a typical transmembrane protein, presumably with a function similar to that of ATG9 in *S*. *cerevisiae*. The regulatory functions of the three autophagy genes *BxATG5*, *BxATG9*, and *BxATG16* and their encoded proteins in autophagy in PWN require further clarification.

Numerous studies have shown that autophagy has a major impact on the growth and development of many organisms, and autophagy genes are reported to play critical roles in the development, reproduction, and pathogenicity of pathogens and insect pests including *Magnaporthe grisea*, *Rhipicephalus* (*Boophilus*) *microplus*, and *Tenebrio molitor* [17,18,19,20]. Therefore, we first studied the effects of three autophagy genes, *BxATG5*, *BxATG9*, and *BxATG16,* on feeding and reproduction in PWN. RNAi technology provides an effective tool to study gene functions and conduct genetic manipulation of plants and animals [35,36], and also has been used to assess the functions of genes in PWN [37,38,39,40]. We successfully silenced *BxATG9* and *BxATG16* using RNAi and found that silencing *BxATG16* significantly reduced the expression level of *BxATG5*. This surprising result indicates that *BxATG16* may positively regulate the expression of *BxATG5*, not only the interaction between their protein products. This effect has not been reported in previous studies of the regulatory mechanisms of autophagy, and merits more detailed and targeted exploration in the future. Silencing *BxATG9* and *BxATG16* did not affect the morphology or motility of PWN, but severely inhibited their feeding and reproduction, in accordance with the results of a study on *BxATG1* and *BxATG8* by Deng et al. [21]. This finding indicates that these autophagy genes are essential to the developmental processes of PWN. However, the potential regulatory mechanisms involving these autophagy genes in PWN require further investigation.

During the early stage of PWD, the defense responses of the host pine trees are stimulated by a large number of pathogenic factors produced by PWN, the main factor being the accumulation of monoterpenes, particularly pinene. α-Pinene is one of the most important volatile monoterpenes [41], and Takeuchi et al. found that release of α-pinene was highly elevated in pine trees infected with PWN [42]. To investigate whether the mechanism by which PWN defend against pine monoterpenes is related to autophagy, we examined the expression levels of the autophagy genes *BxATG5*, *BxATG9*, and *BxATG16* in the nematodes during α-pinene stress. The results showed that the expression of these three autophagy genes was significantly upregulated when PWN were stressed with α-pinene for 4 h, suggesting that autophagy in PWN likely plays an important role in detoxification and repair in response to the large amount of monoterpenes accumulated in pine trees during the early stage of PWD.

Plants have evolved sophisticated mechanisms utilizing compartmentalized production of ROS to modulate defense responses against pathogen attack, as excess ROS has a destructive effect on pathogens [29]. Research has shown that oxidative stress has a destructive effect on macromolecular substances such as proteins, fats, and DNA in human cells, and autophagy can degrade these damaged macromolecular substances to help cells resist oxidative stress [43]. Human lung embryo fibroblasts that were prematurely debilitated with H_2_O_2_ employed autophagy to avoid apoptosis and maintain survival [44]. H_2_O_2_ is the most abundant ROS in pine trees. Both pine trees and PWN produce large amounts of ROS during the early infection stage of PWD, and the H_2_O_2_ content in pine trees increases sharply [30,45]. Therefore, we examined the autophagy response of PWN under oxidative stress and analyzed changes in the expression of the autophagy genes *BxATG5*, *BxATG9*, and *BxATG16* when the nematodes were stressed with H_2_O_2_. The results showed that these three autophagy genes were highly expressed at different times during stress, suggesting that PWN may resist oxidative stress by increasing autophagic activity, and that the investigated autophagy genes regulate different processes of autophagy. Scherz-Shouval et al. also showed that the accumulation of ROS stimulated autophagy in mammalian cells [46].

The expression levels of some autophagy genes in PWN were reported to increase significantly within 24 h after inoculation of pine trees with nematodes [23], and autophagy may help nematodes adapt to the initial defense response of pine trees. To investigate the role of autophagy in the development of PWD, we examined the expression patterns of the autophagy genes *BxATG5*, *BxATG9*, and *BxATG16* in PWN during different stages of PWD. The results showed that the expression levels of these three autophagy genes in PWN all increased to varying degrees from before the onset of symptoms in *P. massoniana* to the early and late stages of symptomatic infection and when the trees died. All three genes were most highly elevated during the late stage. These results indicate that autophagy in PWN appears to play a role in the overall process of PWD, but the importance of this remains to be tested. Atg26-mediated pexophagy was proved to be required for host invasion by the plant pathogenic fungus *Colletotrichum orbiculare* [47]. Before symptoms appeared in pines, nematodes may have enhanced their autophagic activity by increasing the expression levels of autophagy genes to resist defensive responses such as pine monoterpenes and ROS production. Deng et al. showed that starvation can induce autophagy in PWN [21]. In the later stage of infection, there were huge amounts of nematodes in each pine tree, and the nutrient levels in the trees were insufficient, thus starvation likely drove the increased expression levels of autophagy genes in PWN. When pine trees infected with nematodes died, blue stain fungus multiplied rapidly in the trees, and the nematodes could feed on the fungus to obtain nutrients [48]; thus autophagy genes were downregulated. This study confirmed that autophagy genes in PWN responded significantly to abiotic and biotic stresses, suggesting that autophagy may be an important process in the defense and pathogenesis of this nematode. However, further research into the role of autophagy in the growth and pathogenesis of PWN remains necessary.

In summary, our study cloned and characterised three novel autophagy genes, *BxATG5*, *BxATG9*, and *BxATG16*, in PWN, and we found that *BxATG16* positively regulated the expression of *BxATG5*. The results are important to better understand the mechanism of autophagy regulation in PWN. In addition, silencing *BxATG9* and *BxATG16* severely inhibited feeding and reproduction of PWN, which indicates that the autophagy genes are essential to the developmental processes of PWN. Moreover, the expression levels of *BxATG5*, *BxATG9*, and *BxATG16* all significantly increased after nematodes were stressed with α-pinene and H_2_O_2_, the main defence substances of pine trees, and during the development of PWD, suggesting that autophagy plays an important role in the defence mechanisms of PWN and pathogenesis of PWD.

## 4. Materials and Methods

### 4.1. Biological Materials and Growth Conditions

The AMA3 strain of PWN, which has been described as highly virulent [49], was isolated from wood chips of infested *Pinus thunbergii* from Maanshan, China. The nematodes were maintained at 25 °C for one week on a necrotrophic fungus, *Botrytis cinerea*, grown on a potato dextrose agar (PDA) plate (containing 7 g potatoes, 0.7 g dextrose, 0.7 g agar, and 30 mL sterile water). Mixed stages of nematodes were harvested with a Baermann funnel [50], and washed with M9 buffer (30 g/L KH_2_PO_4_, 60 g/L K_2_HPO_4_, and 50 g/L NaCl) prior to use in experiments. Two-year-old *P. massoniana* seedlings were obtained from the greenhouse at Nanjing Forestry University (Nanjing, China) and cultivated at temperatures ranging from 28 to 32 °C and relative humidity of 65%–75%.

### 4.2. PWN RNA Extraction and cDNA Synthesis

The total RNA of the harvested nematodes was extracted using TRIzol reagent (Invitrogen, Carlsbad, CA, USA). The RNA was examined through electrophoresis on a 1% agarose gel and quantified using a Nanodrop 2000C spectrophotometer (Thermo Fisher Scientific, Waltham, MA, USA). cDNA was synthesized from 2 μg of total RNA using TransScript II One-Step gDNA Removal and cDNA Synthesis SuperMix (TransGen Biotech, Beijing, China) according to the manufacturer’s instructions.

### 4.3. Cloning and Sequencing of Coding Sequences for the Autophagy Genes BxATG5, BxATG9, and BxATG16 from PWN

Transcriptome data for PWN were downloaded from the WormBase website, and the CDS of the possible autophagy genes *ATG5*, *ATG9*, and *ATG16* were queried and identified based on their annotation files and transcriptome sequences. Gene-specific primers were designed to target these sequences (Table 1). PCR was performed using PWN cDNA as a template. The PCR conditions were: 98 °C for 2 min followed by 35 cycles of denaturation at 98 °C for 10 s, annealing at 50 °C for 30 s, and extension at 72 °C for 45 s. The final extension step was 72 °C for 5 min. The amplified PCR products were confirmed through electrophoresis on 1% agarose gels and were purified according to the protocol for the Gel Extraction Kit (Axygen, Hangzhou, China). Purified PCR products were cloned into the pClone007 Blunt Simple Vector (TsingKe, Beijing, China), which was used to transform competent cells of *Escherichia coli* Trans1-T1 (TransGen Biotech, Beijing, China). The *E. coli* culture was then incubated overnight at 37 °C on Luria-Bertani (LB) plates containing ampicillin. Positive transformants were analyzed through PCR using the primers M13F (−47) and M13R (−48; Table 1). Once transformed clones were identified, a fresh bacterial suspension from each clone was submitted to Nanjing GenScript (Nanjing, China) for sequencing.

### 4.4. Bioinformatic Analysis of the Autophagy Genes BxATG5, BxATG9, and BxATG16 from PWN

After obtaining accurate CDS of autophagy genes *BxATG5*, *BxATG9*, and *BxATG16*, blastx, amino acid prediction, and homology analysis were performed using NCBI tools, and phylogenetic trees of these three proteins encoded by autophagy genes were constructed using MEGA7 with the maximum likelihood method. The molecular weight, isoelectric point, acidity, and hydrophobicity of each autophagy protein were predicted using the bioinformatics website ExPASy, and the signal peptide, transmembrane structure and tertiary structure were analyzed using the ExPASy (SIB Swiss Institute of Bioinformatics, Zurich, Switzerland), SignalP4.1 (Center for Biological Sequence Analysis, Technical University of Denmark, Lyngby Denmark), TMHMM (Center for Biological Sequence Analysis, Technical University of Denmark, Lyngby Denmark), and SWISS-MODEL (Torsten Schwede, Swiss Institute of Bioinformatics, Basel, Switzerland) programs.

### 4.5. Interference with the Autophagy Genes BxATG9 and BxATG16 Using Double-Stranded RNA

Double-stranded RNAs (dsRNAs) corresponding to the autophagy genes *BxATG9* and *BxATG16* and the negative control gene, green fluorescent protein (GFP), were synthesized using the MEGscript RNAi Kit (Ambion Inc., Austin, TX, USA) with specific primers containing the T7 promoter. Nematodes (a mixture of juveniles and adults) were soaked in BxATG9 dsRNA (dsBxATG9), BxATG9 dsRNA (dsBxATG16), GFP dsRNA (dsGFP), or dsRNA-free (CK) solutions (800 ng/µL), and incubated at 25 °C in a shaking incubator with a rotation rate of 180 rpm for 48 h [51]. Each treatment included three replicates. Samples from each treatment were washed thoroughly with double distilled H_2_O (ddH_2_O) three times after soaking before use in subsequent experiments.

### 4.6. qRT-PCR

The silencing efficiency of the autophagy genes *BxATG9* and *BxATG16* was evaluated through qRT-PCR with specific primers (Table 1). qRT-PCR was carried out using SYBR Green Master Mix (Vazyme, Nanjing, China). The *Actin* gene of PWN was used as an internal control with the primers listed in Table 1. Relative expression levels (fold) were determined using ABI Prism 7500 software (Applied Biosystems, Foster City, CA, USA) and the 2^−ΔΔCt^ method [52]. qRT-PCR was conducted with three biological and technical replicates.

### 4.7. Analysis of Feeding and Reproduction of PWN after RNAi

The effects of autophagy gene RNAi on the feeding and reproductive rates of PWN were tested using PDA plates inoculated with *B. cinerea* at 25 °C for five days. About 500 nematodes from various treatments (CK, dsGFP, dsBxATG9, and dsBxATG16) were selected, transferred onto PDA plates with *B. cinerea,* and cultured at 25 °C. Three biological replicates were used for each treatment. The feeding area of the nematodes was photographed daily and the color of the media darkened as the feeding rate of the nematodes increased. Subsequently, the nematodes were isolated from the PDA plates using the Baermann funnel method and counted under an optical microscope (Leica DM500, Leica Microsystems, Heerbrugg, Switzerland).

### 4.8. Analysis of Expression Levels of BxATG5, BxATG9, and BxATG16 in PWN under Abiotic Stress and at Different Stages of PWD

Four samples of 10,000 nematodes each were soaked in 1 mL of 0.5% Triton X-100, 56.33 mg/mL α-pinene dissolved in 0.5% Triton X-100, ddH_2_O, and 15 mM H_2_O_2_ for 1, 2, 3, and 4 h, respectively [23,28]. Triton X-100 and ddH_2_O were used as controls. Approximately 10,000 nematodes were inoculated into two-year-old *P. massoniana* seedlings, and nematodes were isolated from the seedlings 6, 12, 28, and 40 days after inoculation using the Baermann funnel method. Three biological replicates were used for each treatment. After washing the nematodes from various treatments, their RNA was extracted and cDNA was synthesized. qRT-PCR was performed to determine the relative gene expression levels (fold) of *BxATG5*, *BxATG9*, and *BxATG16* with specific primers (Table 1).

### 4.9. Statistical Analysis

Three replicates were used for each assay. The mean and standard deviation (SD) of each set of three replicates were calculated using Microsoft Excel software (Microsoft Corp., Redmond, WA, USA). Paired Student’s *t*-tests were conducted using SPSS software (ver. 17.0; IBM China Company Ltd., Beijing, China) to compare each treatment with the control. * *p* < 0.05 and ** *p* < 0.01 indicate statistically significant differences and extremely significant differences, respectively.

## Figures and Tables

**Figure 1 ijms-20-03769-f001:**
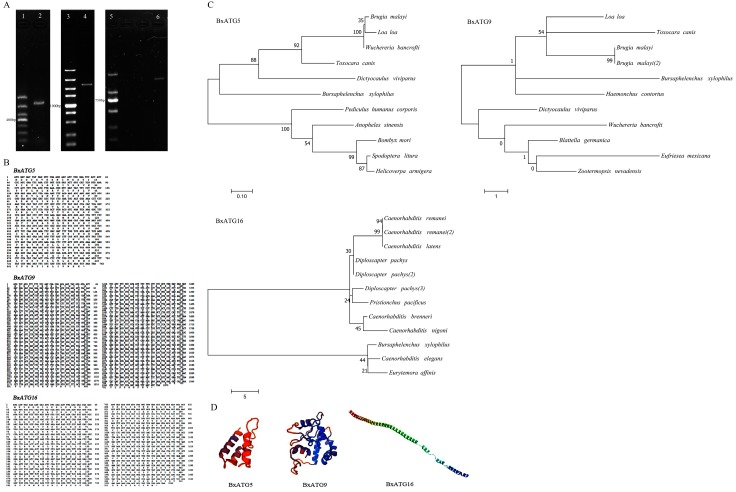
Cloning and sequence analysis of the autophagy genes BxATG5, BxATG9, and BxATG16 in pine wood nematodes (PWN). (**A**) Agarose gel electrophoresis of the coding sequences of BxATG5, BxATG9, and BxATG16. 1: DL1000 DNA Marker; 2. BxATG5; 3. DL5000 DNA Marker; 4. BxATG9; 5. DL2000 DNA Marker; and 6. BxATG16. (**B**) Coding sequences and deduced amino acid sequences of BxATG5, BxATG9, and BxATG16. (**C**) Phylogenetic relationships of the autophagy proteins BxATG5, BxATG9, and BxATG16. (**D**) Tertiary structure models of BxATG5, BxATG9, and BxATG16.

**Figure 2 ijms-20-03769-f002:**
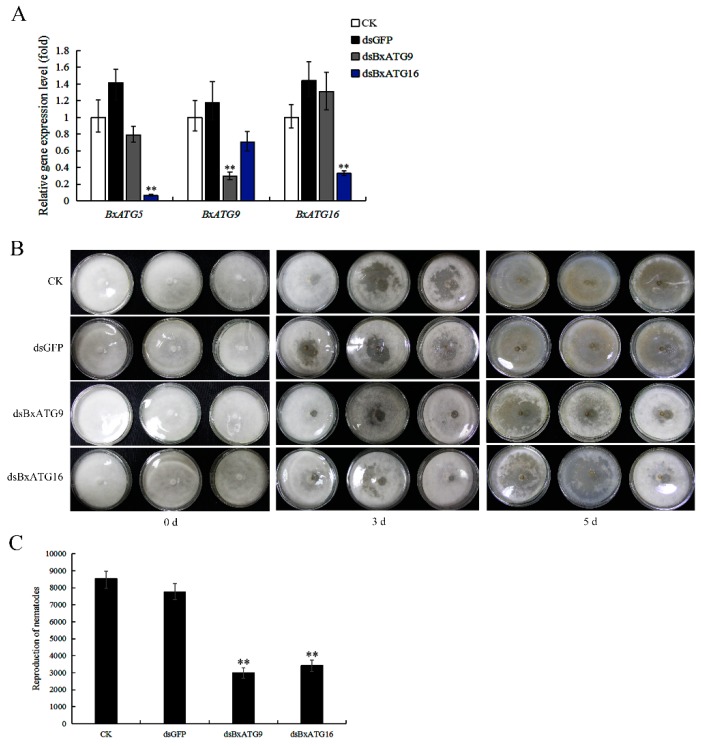
Effects of silencing the autophagy genes *BxATG9*, and *BxATG16* on feeding and reproduction in PWN. (**A**) Silencing efficiency of *BxATG9* and *BxATG16* after RNAi with dsBxATG9 and dsBxATG16 and the effects of RNAi on the expression of other autophagy genes in PWN. PWN soaked in dsRNA-free solution (CK) and dsGFP were used as controls. The expression level of CK was set to 100%. (**B**) Effects of silencing *BxATG9* and *BxATG16* on the feeding rates of PWN on potato dextrose agar (PDA) plates with *Botrytis cinerea*. The media darkened with increased feeding. (**C**) Reproductive assay. Nematodes from different treatments were isolated from PDA plates using the Baermann funnel method and counted. Data are presented as the mean ± SD. ** *p* < 0.01.

**Figure 3 ijms-20-03769-f003:**
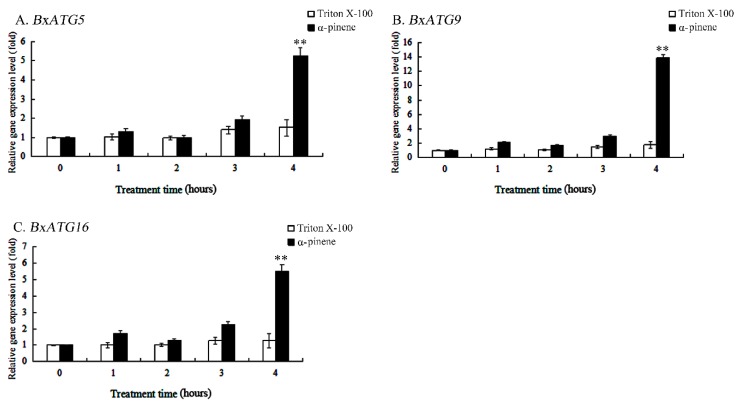
Relative expression levels of the autophagy genes *BxATG5*, *BxATG9*, and *BxATG16* in PWN under α-pinene stress. Triton X-100 was used as the control. (**A**) *BxATG5*. (**B**) *BxATG9*. (**C**) *BxATG16*. Data are presented as the mean ± SD. ** *p* < 0.01.

**Figure 4 ijms-20-03769-f004:**
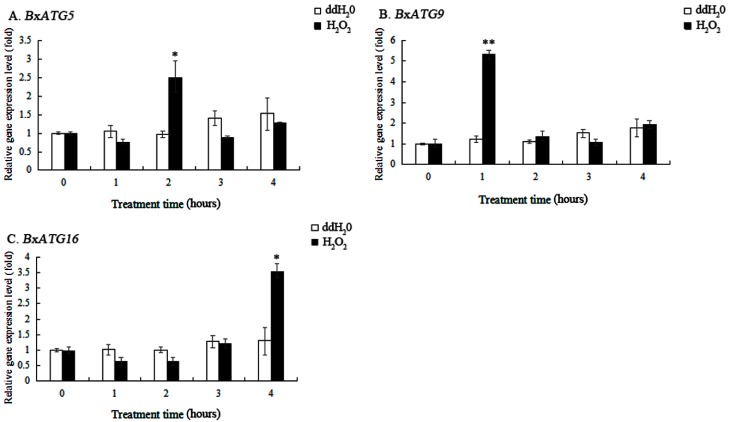
Relative expression levels of the autophagy genes *BxATG5*, *BxATG9*, and *BxATG16* in PWN under H_2_O_2_ stress. ddH_2_O was used as the control. (**A**) *BxATG5*. (**B**) *BxATG9*. (**C**) *BxATG16*. Data are presented as the mean ± SD. * *p* < 0.05 and ** *p* < 0.01.

**Figure 5 ijms-20-03769-f005:**
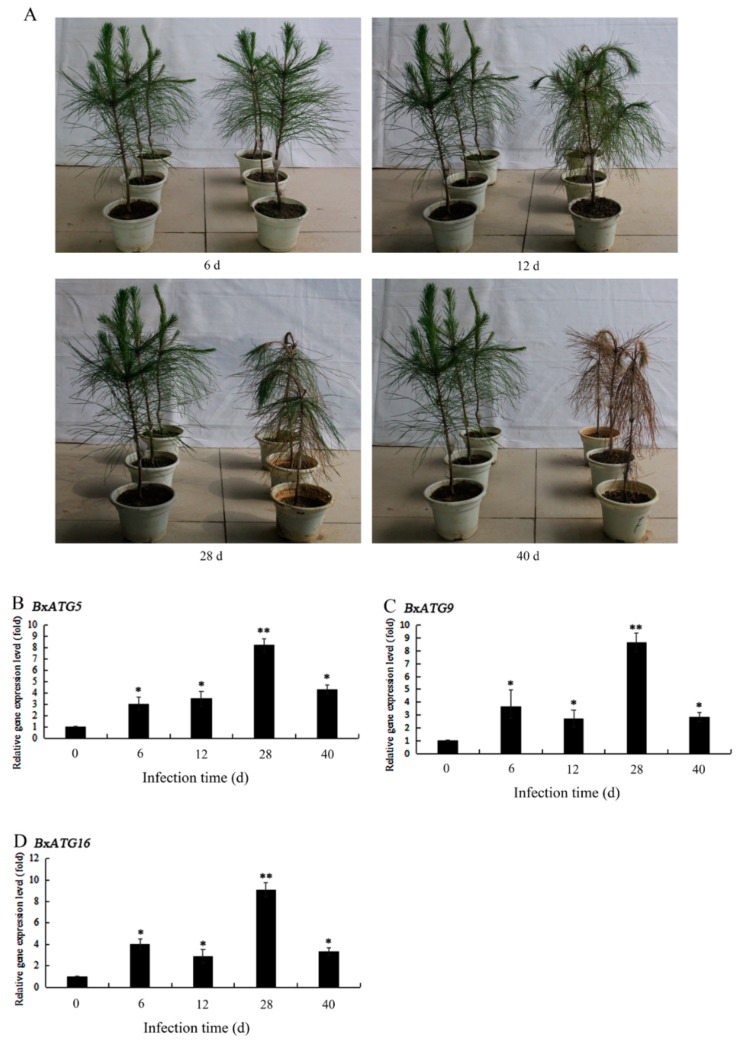
Relative expression levels of the autophagy genes *BxATG5*, *BxATG9*, and *BxATG16* in PWN during the development of pine wilt disease. (**A**) Symptoms in *Pinus massoniana* after inoculation with PWN. (**B**) *BxATG5*. (**C**) *BxATG9*. (**D**) *BxATG16*. Data are presented as the mean ± SD. * *p* < 0.05 and ** *p* < 0.01.

**Table 1 ijms-20-03769-t001:** PCR primers used in the study.

Primers	Sequence (5′ to 3′)
BxATG5-F	ATGGATGACTATGAGGTCCG
BxATG5-R	TTATTTAGATTTGAAGATGAGATGG
BxATG9-F	ATGTCTATGTTTTTTTCCAATCG
BxATG9-R	TTAGACATTAAACCCGCCTG
BxATG16-F	ATGACTTACAGAGATGACATCCT
BxATG16-R	TTACATCCACAAACAGGCA
M13F (−47)	CGCCAGGGTTTTCCCAGTCACGAC
M13R (−48)	AGCGGATAACAATTTCACACAGGA
iBxATG9-F	TAATACGACTCACTATAGGGAGAATTACAATGAGTTGGATCACG
iBxATG9-R	TAATACGACTCACTATAGGGAGATGATCGGCATAACAGGG
iBxATG16-F	TAATACGACTCACTATAGGGAGAAATGATGAACTTTTGGCTCT
iBxATG16-R	TAATACGACTCACTATAGGGAGACTTCCGGGAGCTTCTTC
iBxATG9-qF	AAGACTGAAGTTGAGACA
iBxATG9-qR	ATTATGGCGAAGATGGAT
iBxATG16-qF	AACTACTAACGAATTGCTAA
iBxATG16-qR	ATCAACACCACTCTTTCT
Actin-F	GCAACACGGAGTTCGTTGTAGA
Actin-R	GTATCGTCACCAACTGGGATGA
qBxATG5-F	CAAACGATGAAACAACTCCTTAT
qBxATG5-R	CGAAGCCGCTGATATTACA
qBxATG9-F	GCGGTCAAGTATATGGAT
qBxATG9-R	ATTATGGCGAAGATGGAT
qBxATG16-F	CCAGTTCTCATTACGATAA
qBxATG16-R	AGTAGTTGACCATCTGTA

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
