# Peer review of "Molecular Characterization and Functional Analysis of Three Autophagy Genes, BxATG5, BxATG9, and BxATG16, in Bursaphelenchus xylophilus"

_ijms, 2019, doi:10.3390/ijms20153769_

Round 1
Reviewer 1 Report
Liu et al have produced a well written, logically structured manuscript which characterises three new genes from Bursaphelenchus xylophilus which shown similarity to know autophagy genes. They perform some basic characterisation of these genes and how they could be regulated in pine wilt disease.
Generally the experiments have been performed well and the authors have been responsible about the extent of the conclusions drawn. There are just a couple of points which could be addressed:
Major points:
1. Results lines 152-155: This experiment has a written description for gene expression levels at 5h, but this is not shown in the graphs. Is there a reason for this?
The authors state “These results indicate that these three autophagy genes in PWN were not sensitive to α-pinene stress, but that the nematodes still upregulate autophagy genes after being stressed for a certain period of time.” I think rather than claim the genes are not sensitive to α-pinene stress, they might be better to state that a-pinene exposure is unlikely to directly affect ATG levels (i.e. no immediate response within 1-2 h), but autophagy may be a later response used by the nematodes, and so the genes are upregulated transiently at a later time point (4 h). Without cell data to show autophagic vesicles, etc, it is difficult to say when autophagy might be activated following α-pinene stress and when it might reduce again and how this might correlate with the gene levels.
2. Discussion, line 283: “All three genes were most highly elevated during the late stage. These results indicated that autophagy in PWN plays an essential role in the overall process of PWD.” While the authors have shown changes in autophagy gene regulation, the experiments to prove that this response is essential have not been done. i.e. they have not used worms with knock-down of these autophagy genes and tested whether they can successfully survive α-pinene and H2O2 stress and if they can successfully infect the trees. Therefore, for the experimentation shown, the authors would be better to state that autophagy appears to play a role in infection, but the importance of this remains to be tested.
Similarly, Discussion, line 295 states that “autophagy is a central process in the defence and pathogenesis of this nematode”, but this should be reworded to reflect that the experiments have not yet shown it is central to defence (e.g. knockdown expts, as described above).
The abstract shows better wording, using “suggesting that autophagy plays an important role”. The other parts of the text should be similarly cautiously worded.
3. In the later stage of infection, there were huge amounts of nematodes in each pine tree, and the nutrient levels in the trees were insufficient, thus starvation likely drove the increased expression levels of autophagy genes in PWN. How do the authors know nutrient levels were insufficient? Is this assumed or tested?
Minor Point:
1. “Autophagy bubbles” – these are more commonly referred to as autophagy vesicles.
Author Response
Point 1: Liu et al have produced a well written, logically structured manuscript which characterises three new genes from Bursaphelenchus xylophilus which shown similarity to know autophagy genes.
Response 1: Thank you very much for your recognition of our manuscript.
Point 2: Results lines 152-155: This experiment has a written description for gene expression levels at 5h, but this is not shown in the graphs. Is there a reason for this?
Response 2: We also think that this part needs to be explained. We did not add the gene expression levels at 5h to the graphs, just to make the horizontal coordinate size of the graphs in this result consistent with those in the latter two results, which will make the article look more comfortable.
Point 3: The authors state “These results indicate that these three autophagy genes in PWN were not sensitive to α-pinene stress, but that the nematodes still upregulate autophagy genes after being stressed for a certain period of time.” I think rather than claim the genes are not sensitive to α-pinene stress, they might be better to state that a-pinene exposure is unlikely to directly affect ATG levels (i.e. no immediate response within 1-2 h), but autophagy may be a later response used by the nematodes, and so the genes are upregulated transiently at a later time point (4 h). Without cell data to show autophagic vesicles, etc, it is difficult to say when autophagy might be activated following α-pinene stress and when it might reduce again and how this might correlate with the gene levels.
Response 3: Revised according to the comment. These results indicate that α-pinene exposure is unlikely to directly affect the expression levels of autophagy genes, but autophagy may be a later response used by the PWN, and so these three autophagy genes are upregulated transiently at a later time point. At this point, the nematodes likely act to protect their cells by increasing the expression of autophagy genes to remove damaged proteins and organelles, meaning that autophagy genes may still play an important role in nematode resistance to resin in pine trees.
Point 4: Discussion, line 283: “All three genes were most highly elevated during the late stage. These results indicated that autophagy in PWN plays an essential role in the overall process of PWD.” While the authors have shown changes in autophagy gene regulation, the experiments to prove that this response is essential have not been done. i.e. they have not used worms with knock-down of these autophagy genes and tested whether they can successfully survive α-pinene and H2O2 stress and if they can successfully infect the trees. Therefore, for the experimentation shown, the authors would be better to state that autophagy appears to play a role in infection, but the importance of this remains to be tested.
Response 4: Revised according to the comment. These results indicate that autophagy in PWN appears to play a role in the overall process of PWD, but the importance of this remains to be tested.
Point 5: Similarly, Discussion, line 295 states that “autophagy is a central process in the defence and pathogenesis of this nematode”, but this should be reworded to reflect that the experiments have not yet shown it is central to defence (e.g. knockdown expts, as described above). The abstract shows better wording, using “suggesting that autophagy plays an important role”. The other parts of the text should be similarly cautiously worded.
Response 5: Revised according to the comment. This study confirmed that autophagy genes in PWN responded significantly to abiotic and biotic stresses, suggesting that autophagy may be an important process in the defence and pathogenesis of this nematode.
Point 6: In the later stage of infection, there were huge amounts of nematodes in each pine tree, and the nutrient levels in the trees were insufficient, thus starvation likely drove the increased expression levels of autophagy genes in PWN. How do the authors know nutrient levels were insufficient? Is this assumed or tested?
Response 6: This is speculated. Yang et al confirmed that PWN fed nutrients from pine xylem cells and most of the cells in the pine trees have died and severe water shortage has occurred in the later stage of pine wilt disease. So we speculated that the nutrition was limited at this time with a large number of nematodes.
Yang, B.J.; Pan, H.Y.; Tang, J.; Wang, Y.Y.; Wang, L.F. Pine wood nematode disease. Beijing, China: Forestry Publishing House, 2003. pp. 40-50.
Point 7: “Autophagy bubbles” – these are more commonly referred to as autophagy vesicles.
Response 7: Revised according to the comment.
Reviewer 2 Report
The authors have provided some interesting insight and results regarding the potential control of the PWN through silencing of specific autophagy genes. Overall, there is no major criticisms to be made. How, in practical terms, do the authors envision the specific control method to be used in the field? Specifics: Line 74: I believe that resin is more appropriate than rosin, which is a more solidified and external product, from resin. Line 90: define CDS. Line 99: Toxocara canis and Brugia malay should be in italic. Line 188: The PWN....
Author Response
Point 1: The authors have provided some interesting insight and results regarding the potential control of the PWN through silencing of specific autophagy genes. Overall, there is no major criticisms to be made. How, in practical terms, do the authors envision the specific control method to be used in the field?
Response 1: Thank you very much for your recognition of our manuscript. Based on our results, in practical terms, adding autophagy inhibitor to nematocides may be more effective in controlling the PWN. But this assumption requires more experimentation to verify, so we did not show it in the manuscript.
Point 2: Specifics: Line 74: I believe that resin is more appropriate than rosin, which is a more solidified and external product, from resin.
Response 2: Revised according to the comment.
Point 3: Line 90: define CDS.
Response 3: Revised according to the comment.
Point 4: Line 99: Toxocara canis and Brugia malay should be in italic.
Response 4: Revised according to the comment.
Point 5: Line 188: The PWN....
Response 5: Revised according to the comment.